# Connecting Dots between Mitochondrial Dysfunction and Depression

**DOI:** 10.3390/biom13040695

**Published:** 2023-04-20

**Authors:** Mehtab Khan, Yann Baussan, Etienne Hebert-Chatelain

**Affiliations:** 1Department of Biology, University of Moncton, Moncton, NB E1A 3E9, Canada; 2Mitochondrial Signaling and Pathophysiology, University of Moncton, Moncton, NB E1A 3E9, Canada

**Keywords:** mitochondria, depression, OXPHOS, ATP

## Abstract

Mitochondria are the prime source of cellular energy, and are also responsible for important processes such as oxidative stress, apoptosis and Ca^2+^ homeostasis. Depression is a psychiatric disease characterized by alteration in the metabolism, neurotransmission and neuroplasticity. In this manuscript, we summarize the recent evidence linking mitochondrial dysfunction to the pathophysiology of depression. Impaired expression of mitochondria-related genes, damage to mitochondrial membrane proteins and lipids, disruption of the electron transport chain, higher oxidative stress, neuroinflammation and apoptosis are all observed in preclinical models of depression and most of these parameters can be altered in the brain of patients with depression. A deeper knowledge of the depression pathophysiology and the identification of phenotypes and biomarkers with respect to mitochondrial dysfunction are needed to help early diagnosis and the development of new treatment strategies for this devastating disorder.

## 1. Introduction

Mitochondria are highly dynamic organelles forming a network which spans throughout the cytosol [1]. Mitochondria are composed of four compartments: the outer and inner mitochondrial membranes (OMM and IMM, respectively), which are separated by the inter-membrane space (IMS), and the mitochondrial matrix surrounded by the IMM [2]. Mitochondrial morphology can be modulated by events of fusion and fission between individual organelles, which are crucial to maintain mitochondrial activity. Fission involves the GTPase dynamin-1-like protein (Drp1) and is required for mitochondrial quality control [3,4], while fusion is mainly driven by mitofusin 1 (Mfn1), mitofusin 2 (Mfn2) and optic atrophy-1 (OPA1) [5,6,7,8], and allows the transfer of mitochondrial proteins, lipids, metabolites and mitochondrial DNA (mtDNA) between individual mitochondria. Mitochondria are the only organelles to possess their own genome, which mostly encodes components of the electron transport chain (ETC) [9]. One of the major roles of mitochondria is ATP production by the ETC and the ATP synthase. The ETC is composed of complexes I, II, III and IV embedded in special regions of the IMM named cristae and two mobile electron carriers, coenzyme Q and cytochrome c. Complexes I, III and IV generate a proton gradient across the IMM which is used by the ATP synthase to generate ATP. This process is called oxidative phosphorylation (OXPHOS). During this process, reactive oxygen species (ROS) are also generated as normal byproducts [10,11,12]. ROS can act as signaling cues but overaccumulation of ROS can lead to the oxidation of proteins and lipids, eventually resulting in autophagy, apoptosis, necrosis and inflammation [12,13,14]. Mitochondria also play an important role in Ca^2+^ clearance, lipid biogenesis, iron-sulfur (Fe-S) clustering and apoptosis [15,16].

Mitochondria are crucial for the brain’s physiology. The brain has among the highest energy needs in the human body and ATP production by mitochondria is thus essential to maintain the brain’s activity [17,18]. Mitochondria-derived ATP is critical for maintaining the Na^+^-K^+^-ATPase activity and, consequently, the membrane potential of neurons which is constantly disturbed by action potentials during a nerve impulse [19]. The capacity of mitochondria to buffer intracellular levels of Ca^2+^ is essential during synaptic transmission [19]. Thus, any defects of mitochondrial functions can lead to brain-related disorders, such as neurodegenerative diseases and neuropsychiatric disorders [20,21,22,23,24]. The glucose metabolism in different brain regions of patients with mood disorders appears disturbed [25,26,27,28]. In preclinical models of depression, the impaired expression of genes encoding mitochondrial proteins, lower activity of ETC components and mitochondrial metabolism, higher oxidative stress and oxidation of mitochondrial structures are all observed [29,30,31].

The aim of this review is to describe the extent of mitochondrial dysfunction observed in depression, from both clinical and preclinical perspectives.

## 2. Neurobiological Basis of Depression

Depression is a common neuropsychiatric disorder, which is widespread across the world, affecting more than 350 million people globally and leading to 1 million deaths by suicide annually [23,30,32]. Since 2008, the World Health Organization listed depression as the third largest cause of economic and disease burden, and it is expected to be ranked first by 2030 [33,34,35,36]. The traditional diagnosis of depression proposes two subtypes, (1) reactive/neurotic depression and (2) endogenous depression, which are based on the presence or absence of stress prior to the onset of depression, respectively [37]. The symptoms and severity vary among individuals, but can be characterized by persistent sadness, low mood or pleasure, decreased energy, altered appetite, deficits in sleep and cognitive capacities, weight loss or gain, decreased social functioning and increased suicidal probability [35,38,39]. Studies have shown that depression can be associated with other metabolic diseases, including cardiovascular and cerebrovascular impairments, autoimmune diseases, diabetes, cancer and a higher mortality rate [35,38,39,40]. Depression often coexists with other psychiatric conditions such as anxiety disorder [41].

Despite widespread preclinical and clinical research studies, the pathophysiology of depression remains poorly understood [42]. The monoamine hypothesis was one of the first dominant theories to explain the pathogenesis of depression. The disturbance of monoamines, such as serotonin and norepinephrine levels, in the brain disrupts the hypothalamus-pituitary-adrenal axis which controls the response to stress, which ultimately leads to depression [43,44,45]. Monoamine oxidase (MAO) is one of the main enzymes in monoamine metabolism and is a potential biomarker of mental disorders [46,47]. Altered MAO activity in the brain disrupts levels of monoamine neurotransmitters such as dopamine, noradrenaline, and serotonin. Defective MAO also impacts mitochondria: it can affect the structure of mitochondrial membranes by activating oxidative stress and increasing the toxic levels of aldehydes and ammonia [46,47]. However, treatment with monoamines seems to have little beneficial effect on the mood of patients. Only 40% of patients respond to treatment with monoamines, and depression-relapse is often observed, although the levels of monoamines are restored a few hours after injections [48,49].

The neurogenesis theory of depression suggests that stress decreases hippocampal neurogenesis, ultimately leading to depression [50]. Decreased hippocampal volume is observed in depressed patients, and decreased hippocampal neurogenesis and neuronal maturation are observed in mice and rats treated with corticosterone, a classic model of depression [30,50,51,52,53,54]. Interestingly, administration of antidepressants can rescue cell proliferation and survival within the hippocampus in animal models [55]. Reelin, an extracellular matrix protein which regulates adult hippocampal neurogenesis and dendritic spine plasticity could also be involved in the pathophysiology of depression. Indeed, injection of corticosterone in rats dampens the expression of reelin in the dentate gyrus of the hippocampus, which can be reversed by treatment with antidepressants [51,56,57].

Neurogenesis is also believed to be an important mechanism under physiological conditions and in brain repair after different injuries such as hypoxia and stroke [58,59]. In animal models of depression, hippocampal cell proliferation and neurogenesis is also altered [60]. Interestingly, the proliferation of neural precursor cells in the subgranular layer of hippocampus decreases, which can be restored by treatment with an antidepressant such as fluoxetine, tranylcypromine, reboxetine and rolipram [61]. Antidepressants, such as selective serotonin reuptake inhibitors and tianeptine, a modified tricyclic antidepressant, increase neurogenesis in the dentate gyrus of mice and non-human primates [55,60,62,63,64]. Similarly, staining with Ki67, which binds onto the proliferating cells, is reduced in the dentate gyrus of post-mortem brains of depressed patients [65,66]. In contrast, the treatment of a depressed patient with antidepressant increases the proliferation of the neural precursor cells in the dentate gyrus as compared to non-treated depressed patients or healthy controls [65]. Many studies have shown that during neuronal development, the mitochondrial biogenesis takes place at a faster rate as neuronal differentiation requires an increased mitochondrial genome and mitochondrial proteins [67,68,69,70,71]. Therefore, it is notable that mitochondrial dysfunction might have an important role in impaired adult hippocampal neurogenesis in depression. However, the neurogenesis hypothesis remains somewhat controversial since depression-like symptoms can occur even when the cell proliferation within the hippocampus is not decreased [72]. Also, antidepressants do not always increase hippocampal neurogenesis in animal models of depression [72,73,74]. In fact, the evidence suggests that depression and the efficacy of antidepressants may be more related to variations in dendritic plasticity and neuronal remodeling than neurogenesis [72,74,75]. Therefore, the role of neuroplasticity might be more important in the pathophysiology of depression. Neuroplasticity includes synaptic and non-synaptic plasticity in response to internal and external stimuli. Synaptic plasticity encompasses axonal and dendrite growth, synaptogenesis, and removal of defective connections between neurons. Mitochondria play an important role in neuroplasticity, and it is well known that impaired neuroplasticity due to mitochondrial stress leads to structural and functional impairment in different regions of the brain of depressed patients [23,76]. Overall, mitochondrial dysfunction might have important roles in impaired adult hippocampal neurogenesis and neuroplasticity in depression.

Depressed patients show pathological alterations in selective brain regions, including limbic (hippocampus, basal ganglia and amygdala) and cortical brain regions. These brain regions are involved in affective and cognitive impairments observed in depression [77]. Notably, neuroimaging studies by magnetic resonance imaging and positron emission tomography revealed metabolic alterations in depressed patients [78,79]. The brain regions metabolically impaired in depressed patients are (i) the cortical areas such as the prefrontal cortex, cingulate cortex, orbital frontal cortex and insula; (ii) the subcortical limbic regions such as the amygdala, hippocampus and the dorsomedial thalamus; and (iii) the basal ganglia and the brain stem region [78,80].

## 3. Models of Depression

The post-mortem brains of depressed patients are a crucial source of information for the pathophysiology of depression. However, it is important to consider that they are associated with important artefacts, such as increased oxidative stress that occurs during and after death [81,82]. The animal models of depression are a widely used alternative to the post-mortem brains of humans despite the fact that no animal model can perfectly mimic human depression due to its multifactorial pathophysiology [83,84]. Nonetheless, the animal models of depression helped improve our understanding of the pathophysiology of depression and its link with mitochondrial dysfunction (see Table 1). Three main criteria must be fulfilled to generate animal models of depression. First, they must have a phenotype similar to humans suffering from depression. Then, they must recapitulate the human physiopathology and should be sensitive to pharmacological or non-pharmacological treatments effective in humans [85,86,87]. The generation of animal models for depression mainly relies on stress exposure [88,89], and several protocols have been developed, based on numerous variables such as the nature of the stress, its severity and exposure parameters to induce and/or measure the depression-like phenotype [90].

The main protocols to induce a depression-like phenotype are exposure to chronic mild stress, social defeat stress and early-life stress [87,110,111,112,113]. These different protocols which are reviewed in [114] induce a depression-like phenotype, which can then be evaluated through various behavioral tests, including the forced swim test, the tail suspension test, the learned helplessness test and the sucrose preference test [111,115,116].

## 4. Mitochondrial Genetics and Depression

Mitochondria are the only organelles to have their own DNA known as mtDNA. The mtDNA encodes 13 polypeptides involved in OXPHOS, 12S and 16S ribosomal RNAs, as well as 22 unique transfer RNAs for protein synthesis [117,118,119]. Several observations link mtDNA and depression. The prevalence rate for depression is 54% in patients with mitochondrial diseases caused by specific mutations of mitochondrial genes [120]. One study revealed that 68% of patients with depression have mtDNA deletions, as compared to 36% of the control patients [121]. Likewise, there are significantly lower mtDNA copy numbers and increased mtDNA oxidative damage in the leukocytes of depressed patients as compared to the control individuals [122]. The resulting damaged mtDNA activates pro-inflammatory cytokines which leads to inflammation, a hallmark of depression [123,124]. Studies in mice and humans also showed that variation in the mtDNA copy numbers is associated with cognitive impairments, which are other common symptoms linked with depression [31,125,126]. The downregulation of DNA repair enzymes such as DNA polymerase gamma (POLG) and 8-oxoguanine-DNA glycosylase 1 (OGG1) in patients with depression suggest that this might be one of the mechanisms responsible for the low mtDNA copy number in depressed patients [127,128].

A number of specific mitochondrial genes have also been linked to depression [31]. Mitochondrial PCR array profiling analysis identified 16 genes differentially expressed in the dorsolateral prefrontal cortex of the post-mortem brains of depressed patients [129]. The identified genes were mainly related to oxidative stress and neuronal ATP levels [129]. Similarly, the *ATP6V1B2*, which encodes a subunit for the vacuolar proton pump ATPase, is upregulated and has shown effects on neurotransmission and receptor-mediated endocytosis that are involved in depression [31,130]. In summary, the defects of an abundance of mtDNA or mitochondrial gene expression support the idea that mitochondrial abnormalities may represent fundamental pathogenic mechanisms in depression.

## 5. Mitochondrial Proteome in Depression

In humans, 1500 different mitochondrial proteins are involved in mitochondrial dynamics, mtDNA maintenance, bioenergetics, mitophagy, import of proteins inside the organelle, and ion channels [131]. Many studies have demonstrated the involvement of the energy metabolism-related proteome in depression. The brains of depressed patients show altered levels of proteins involved in many metabolic pathways including OXPHOS, pyruvate metabolism and the tricarboxylic acid cycle [132]. A proteomic study on the post-mortem brains of depressed patients showed that 21% of mitochondrial proteins have altered levels [133,134]. Another work showed that 20 subunits of the ETC complexes were increased in the post-mortem human brains of depressed patients [81]. Similar findings were observed in animal models of depression. The congenic C57BL/6NTac mutant mouse model of depression shows dysregulation of metabolism and altered levels of OXPHOS protein in the hippocampus, which is the main brain region with reduced neuroplasticity in both human and mice models of depression [135,136]. Similarly, increased levels of ETC complex I and IV, cytochrome c and ATP synthase were also observed in the dorsolateral prefrontal, anterior cingulate and parieto-occipital cortices of depressed patients [81,137,138,139,140,141]. Carbonic anhydrase and aldolase c are also increased in the frontal cortex and the anterior cingulate cortex of depressed patients [137,142].

The antidepressant fluoxetine is part of the initial treatment for depression [143]. Fluoxetine seems to affect levels of cytosolic and mitochondrial proteins differently. In the cytosol, 23 proteins were upregulated whereas 60 were downregulated upon treatment with fluoxetine. In mitochondria, 60 proteins were upregulated and 3 were downregulated upon the same treatment [144], suggesting that this antidepressant could treat depression through its action on the metabolism-related proteome.

Overall, the mitochondrial proteome is affected in depressed patients, raising the possibility of developing mitochondrial biomarkers to follow the etiology of depression or to develop new treatments.

## 6. OXPHOS and ATP Production by Mitochondria in Depression

Mitochondria produce most of the ATP used by cells [81,145]. Preclinical and clinical studies have shown that, levels of neurometabolites, including ATP, are altered in depressed patients [122,123,124,146]. For instance, magnetic resonance spectroscopy has shown altered levels of phosphocreatine (PCr), N-acetyl-aspartate, adenosine diphosphate (ADP) and ATP in depressed patients [121,147]. Decreased mitochondrial ATP production was also observed in the muscles of depressed patients [121]. A decreased activity of ETC complexes I+III and II+III was also reported in the muscles of depressed patients compared to the control [121], suggesting that mitochondrial dysfunction linked to depression is not limited to the brain. Chronic mild stress can induce depression-like behavior, such as reduced sucrose preference and body weight, with the increased immobility time in the tail suspension test, leads to lower mitochondrial respiration, damaged mitochondrial ultrastructure, and mitochondrial depolarization in the hypothalamus, cortex and hippocampus in mice [102,148,149]. Interestingly, the antidepressant fluoxetine can restore sucrose preference, body weight, ATP synthesis and the respiratory control ratio (which is a quality index for OXPHOS) in the raphe nucleus in the chronic stress model in rodents [150]. Overall, these studies support the importance of appropriate ATP production by mitochondria in the pathophysiology of depression.

## 7. Oxidative Stress in Depression

Mitochondria are important sources of ROS, which play major roles in cellular physiology and signaling [151]. The reduction of O_2_ into H_2_O during OXPHOS can be incomplete and generate superoxide anions, which can then be converted into a hydroxyl radical and H_2_O_2_ [152]. ROS can be scavenged by the antioxidant system composed of catalase, superoxide dismutase, glutathione peroxidase and thioredoxin [153]. When the production of ROS exceeds the scavenging capacity of the antioxidant system, it can cause damage to proteins, lipids and DNA, including mtDNA [149,154]. Both preclinical and clinical studies reported mitochondrial dysfunctions linked to increased oxidative stress in depression [29,30,155,156,157]. Post-mortem analyses showed alterations of the complex I subunits NDUFV1, NDUFV2 and NDUFS1 and increased oxidative damage in the cerebellum of depressed patients [158]. A decreased level of antioxidant enzymes localized within mitochondria such as manganese superoxide dismutase was also reported in depressed patients [159,160]. Also, adult male rats stressed by immobilization for 21 days have reduced levels of the antioxidant glutathione with increased lipid peroxidation and levels of nitric oxide (NO) within the brain [161]. Decreased levels of glutathione and increased levels of superoxide, NO and lipid hydroperoxides are also observed upon olfactory bulbectomy, another model of depression in mice and rats [162,163]. Chronic mild stress during 40 days decreases sucrose preference, and inhibits the activity of ETC complexes I, III and IV in the cerebral cortex and cerebellum (Table 1) [101]. Interestingly, in the mouse model of high-anxiety, a common comorbidity of depression, the ETC components have reduced enzymatic activity, and increased ROS levels leading to lipid peroxidation and cell death [164]. Finally, the meta-analysis of 23 published studies suggests that markers of oxidative stress increase with the progression of depression [156,157]. Overall, numerous findings suggest that oxidative stress mediated by mitochondria dysfunction is linked to depression. Nevertheless, additional studies are required to elucidate and understand their mechanistic links with the symptoms of depression [165].

## 8. Calcium Homeostasis and Depression

Mitochondria play important roles in Ca^2+^ signaling as modulators, buffers and sensors of Ca^2+^ intracellular levels [166]. Ca^2+^ is imported through the OMM via the voltage-dependent anion channel and across the IMM through the mitochondrial Ca^2+^ uniporter [167]. The uptake of Ca^2+^ inside the mitochondrion has a significant impact on energy production, neuronal excitability and cellular death [168,169,170]. However, a Ca^2+^ overload within the mitochondria results in mitochondrial depolarization and the inhibition of OXPHOS, mitochondrial swelling, IMM remodeling, opening of the mitochondrial permeability transition pore, release of cytochrome c, activation of caspases and ultimately apoptosis [112,146,171,172]. The dysregulation of mitochondrial Ca^2+^ homeostasis appears to be involved in the pathophysiology of depression. Genome-wide association studies (GWAS) have identified *Cacna1c* as a candidate risk gene for multiple neuropsychiatric disorders, including bipolar disorder, schizophrenia and depression [173]. *Cacna1c* encodes the pore-forming α1C subunit of the L-type Ca^2+^ channel CaV1.2, which represents the major L-type voltage-gated calcium channel in the brain. CaV1.2 channels are critical modulators of many cellular processes involved in the progression of depression. CaV1.2-dependent gene expression plays an important role in neuronal plasticity, dendritic development and cell survival, suggesting that perturbation in CaV1.2 signaling might lead to depressive phenotypes [174,175,176]. The knockdown of CaV1.2 in rats induces an anti-depressive phenotype as assessed by the tail suspension, forced swim and sucrose preference tests, suggesting that loss of CaV1.2 regulates depressive-like behaviors [173,177,178,179]. Similarly, the knockdown of *Cacna1c* in neuronal HT22 cells, protects mitochondrial morphology, the mitochondrial membrane potential, ATP production and calcium homeostasis from glutamate excitotoxicity. The knockdown of *Cacna1c* also reduces the glutamate-induced increase of mitochondrial ROS production, intramitochondrial calcium influx and cell death in HT22 cells [180]. Thus, Ca^2+^ homeostasis appears to play an important role in many psychiatric disorders, including depression.

## 9. Inflammation and Mitochondria in Depression

Inflammation is one of the main processes involved in depression. Several studies have found dysregulation of both the innate and adaptive immune system in depressed patients [181,182]. Chronic psychological or physiological insults result in the activation of many inflammatory responses, including higher levels of circulating pro-inflammatory cytokines and lower levels of anti-inflammatory cytokines [183]. Activation of the pro-inflammatory cytokines interferon-γ, interleukin (IL)-2, 1β, IL-6 and tumor necrosis factor-α (TNF-α) are observed in depressed patients [184,185,186]. Furthermore, reduction in the anti-inflammatory cytokines IL-4 and IL-10 were observed in depressed patients [187], indicating an imbalance between pro-inflammatory and anti-inflammatory cytokines.

Various protocols, including immobilization, physical restraint, psychological stress, open field stress and inescapable shock, known to induce depression-like phenotype in rodents, increase the levels of pro-inflammatory cytokines and NO within the brain and plasma [188,189,190]. Conversely, mice centrally injected with the pro-inflammatory cytokines TNF-α or IL-1β show depressive-like behaviors, such as increased immobility in the forced swim and tail suspension tests, as well as decreased sucrose preference [88,190]. Interestingly, TNF-α impairs the mitochondrial oxidative metabolism as a result of increased ROS production in animal models [191,192,193]. TNF-α has an inhibitory effect on ETC complex IV, leading to a decreased mitochondrial membrane potential and ATP levels [23,194]. Injection of lipopolysaccharide (LPS) induces a strong immune response and secretion of pro-inflammatory cytokines, which results in depression-like behavior in different tests, such as the sucrose preference, the tail suspension and the forced swim tests [195]. Interestingly, LPS-treated mice also have increased mitochondrial production of superoxide and lower ATP production with a blunted mitochondrial membrane potential in the hippocampus [196]. Administration of the antioxidant resveratrol reverses mitochondrial dysfunction and depression-like behaviors in LPS-treated mice [196]. LPS also reduces sucrose preference and increases the mRNA levels of subunits 1 and 3 of complex IV in the prefrontal cortex of rats [197]. Injection of dinitrobenzene sulfonic acid to mimic colitis and gut inflammation induces depression-like phenotype using the same behavioral tests, together with decreased levels of reduced glutathione and ATP, but increased levels of ROS in the hippocampus [198]. Overall, multiple findings support the hypothesis that inflammation and mitochondrial dysfunction are linked in the pathophysiology of depression.

## 10. Conclusions

The exact biological mechanisms underlying depression are still not perfectly elucidated. However, the evidence that is emerging suggests that mitochondrial dysfunction is involved in various psychiatric disorders, including depression. In this manuscript, we have highlighted the potential relevance of mitochondrial dysfunction in the pathophysiology of depression. In depression, mitochondrial dysfunction in terms of altered genetics and mtDNA, expression of mitochondrial proteins, OXPHOS and ATP production and oxidative stress can lead to apoptosis and inflammation, decreased neurogenesis and transmission among neural circuits in the cortex, hippocampus and striatum. Improving the mitochondrial functions could thus prevent or alleviate depression-like symptoms. More work focusing on the mechanistic links between mitochondrial dysfunction and depression may become an important avenue towards the development of new treatments against depression.

## Figures and Tables

**Table 1 biomolecules-13-00695-t001:** Animal models of depression show various signs of mitochondrial dysfunction.

Depression Protocol	Depression-like Behavior	Mitochondrial Alteration	Ref.
Prenatal stress	Not analyzed	Different expression of TOM70 and ATP5F1C	[91]
Maternal separation	Increased immobilization time in the forced swim test	Upregulation of GLUD1, ATP5F1A and ATP5PD, and downregulation of IDH3A	[92,93]
	Reduced appetite, anhedonia	Downregulation of fumarate hydratase and IDH; upregulation of citrate synthase	[94,95]
	Not analyzed	Downregulation of ATP5F1B, complex III subunit I, PDHE1-B	[96]
Chronic mild stress	Anhedonia	Modified expression of Hsp60	[97]
	Anhedonia	Upregulation of ATP5F1C, IDH3B, complex I 75 kDa subunit	[98]
	Anhedonia	Upregulation of NDUFB7, COX5A, COX5B	[99]
	Anhedonia	Downregulation of MDH	[100]
	Anhedonia	Upregulation of COXVa, downregulation of IDH3A and ACO2	[101]
	Anhedonia, reduced weight and decreased locomotion	Downregulation of OGDH, NDUFS3	[102]
	Anhedonia, no weight gain	Inhibition of Complex I, III and IV activities in cerebral cortex and cerebellum but not in hippocampus, prefrontal cortex and striatum	[103]
	Anhedonia, reduced body weight, increased immobility in the tail suspension test	Reduced O_2_ consumption, mitochondrial depolarization in hippocampus, cortex and hypothalamus	[104]
Unpredictable chronic mild stress	Anhedonia, increased immobilization time in the forced swim test	Upregulation of NDUFA4, COXVIb-1, complex III subunit 1, MDH; downregulation of SDHA and SOD2	[105]
	Anhedonia, increased immobilization time in the forced swim test	Downregulation of MFN1, MFN2 and DOD2; reduced activity of complex I and IV, mitochondrial depolarization	[106]
Social defeat stress	Reduced locomotion in the open field test	Upregulation of ACO1 and GDH; downregulation of MDH and GRP75	[107]
Learned helplessness	Reduced active avoidance	Downregulation of ATP5F1A, fumarate hydratase, HSP70, PDHE1-B, EF-Tu, GDH, MDH and aconitase	[108,109]

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
