# Peer review of "Connecting Dots between Mitochondrial Dysfunction and Depression"

_biomolecules, 2023, doi:10.3390/biom13040695_

Round 1

Reviewer 1 Report

The paper entitled “Connecting Dots Between Mitochondrial Dysfunction And Depression” is well written and provides an interesting information about the possible association between mitocondrial dysfunction and depression, due to oxidative stress by one hand, and on the other hand because of hippocampal cell neurogenesis and proliferation alteration. Likewise, the disturbance of monoamines stands out as the main theory of the pathogenesis of depression.

Apart from considering the appropriate order in the presentation of the sections and their interest, I would like to make some comments regarding the content of your work.

First of all, I consider that depression in animal models is not quite equiparable as the depression in humans, in which there should be take into account that there are differences in both types of depression: reactive depression, due to a cause that origins that state, due to physical or mental stress, that it may be resolved when the cause dissapears, and the endogenous depression, in which there is no a justified cause, but the patients feel bad, and the resolution is generally longer. This type of depression is more difficult to improve.

In addition, I consider that the changes relative to electron transport chain (ETC) complexes in post-mortem human brains (line 219 and 264) aren´t very relevant. In fact, the tissues already altered and increase oxidative stress previously to the death.

Moreover, it is not clear if depression is the origin of other complicacions such as cardio and cerebrovascular diseases, diabetes, cancer (line 68), or precisely all these disorders begin firstly and after can coexist with depression and other psychiatric conditions as anxiety disorder.

Finally, I would like to add that there is not any reference in the paper to support that depression is charecterized by “neuronal apoptosis” (line 12), between others alterations.

Minor request:

-References in the text: write them between square brackets.

-Write numbers before the principal paragraphs: 1. Introduction; 2. Neurobiological basis of depression; 3. Animal models of depression…

-Line 42: remove the comma before "autophagy".

-Add final point in lines: 49 (before “The capacity”); line 50 (before “Thus”); 56; 127 (after “depression”); 213; 278.

-Line 129: Undo bold endpoint.

-Line 59, 134, 210, 237, 255, 281: remove two points after the titles.

-Write gene names in cursive.

-Line 134: remove the capital letters in the title.

-Line 237: write “OXPHOS and ATP” in capital letters.

-Line 270: remove “nitric oxide” before “NO”-it was defined previously.

Author Response

Comment #1-1: First of all, I consider that depression in animal models is not quite equiparable as the depression in humans, in which there should be take into account that there are differences in both types of depression: reactive depression, due to a cause that origins that state, due to physical or mental stress, that it may be resolved when the cause dissapears, and the endogenous depression, in which there is no a justified cause, but the patients feel bad, and the resolution is generally longer. This type of depression is more difficult to improve.

Response:

The reviewer has pointed out a very critical point. Manuscript was modified accordingly (lines 63-64,128-129).

Comment #1-2: The changes relative to electron transport chain (ETC) complexes in post-mortem human brains (line 219 and 264) aren´t very relevant. In fact, the tissues already altered and increase oxidative stress previously to the death.

Response:

We agree that post-mortem samples can be associated with artefacts. Nevertheless, studying post-mortem samples is a useful approach to detect alterations in human samples. These biases are now acknowledge in the manuscript (lines 209-212).

Comment #1-3: Moreover, it is not clear if depression is the origin of other complicacions such as cardio and cerebrovascular diseases, diabetes, cancer (line 68), or precisely all these disorders begin firstly and after can coexist with depression and other psychiatric conditions as anxiety disorder.

Response:

We agree and the manuscript was modified accordingly (lines67-68).

Comment #1-4: There is not any reference in the paper to support that depression is charecterized by “neuronal apoptosis” (line 12), between others alterations.

Response: Reference to neuronal apoptosis was removed in the abstract since there are no work showing link between mitochondria-dependent apoptosis in depression.

Comment #1-5:

-References in the text: write them between square brackets.

-Write numbers before the principal paragraphs: 1. Introduction; 2. Neurobiological basis of depression; 3. Animal models of depression…

-Line 42: remove the comma before “autophagy”.

-Add final point in lines: 49 (before “The capacity”); line 50 (before “Thus”); 56; 127 (after “depression”); 213; 278.

-Line 129: Undo bold endpoint.

-Line 59, 134, 210, 237, 255, 281: remove two points after the titles.

-Write gene names in cursive.

-Line 134: remove the capital letters in the title.

-Line 237: write “OXPHOS and ATP” in capital letters.

-Line 270: remove “nitric oxide” before “NO”-it was defined previously.

Response:

The manuscript was edited accordingly.

Reviewer 2 Report

Khan et al presented in the current review a comprehensive overview about mitochondrial function and depression. They cited most important literature in the field and provided an good overview about the knowledge in the field.

I have some minor comments:

(I) They used subsections but forgot to link the known literature from each subsection with each other. Now it sounds that all parts stands for its own but are not linked. It would be better to link the sections and the knowledge and it would be very excellent if the authors can provide an illustrating model combining the knowledge about mitochondria (dys)function in depression. 

(II) They describe a lot about depression mouse models and behavioural tests which can be used. There is no need to describe the behavioural tests from animal models in that detail.

(III) The provided in table 1 an overview about mitochondrial dysfunction found in depression models. But there is no description about that in the main text und there is also no link of that data to human data. This table stands completely alone and is not linked to the review at all.

Author Response

Reviewer 2:

Comment #2-1: They used subsections but forgot to link the known literature from each subsection with each other. Now it sounds that all parts stands for its own but are not linked. It would be better to link the sections and the knowledge and it would be very excellent if the authors can provide an illustrating model combining the knowledge about mitochondria (dys)function in depression.

Response: We now include a recapitulating graphical abstract illustrating the different mitochondrial alterations presented in the manuscript.

Comment #2-2: They describe a lot about depression mouse models and behavioural tests which can be used. There is no need to describe the behavioural tests from animal models in that detail.

Response: Respectfully, we disagree with the Reviewer. We believe this section is of interest for non-specialists, as it is rarely explained in other manuscripts.

Comment #2-3: The provided in table 1 an overview about mitochondrial dysfunction found in depression models. But there is no description about that in the main text und there is also no link of that data to human data. This table stands completely alone and is not linked to the review at all.

Response: The table is presenting the specific types of mitochondrial alteration related to the different animal models of depression. Thus, this table is not relevant for human data. The table is now cited in the main text.